# Activation of Piezo1 Increases Na,K-ATPase-Mediated Ion Transport in Mouse Lens

**DOI:** 10.3390/ijms232112870

**Published:** 2022-10-25

**Authors:** Mohammad Shahidullah, Joaquin Lopez Rosales, Nicholas Delamere

**Affiliations:** 1Department of Physiology, University of Arizona, 1501 N Campbell Avenue, Tucson, AZ 85724, USA; 2Department of Ophthalmology and Vision Science, University of Arizona, 1501 N Campbell Avenue, Tucson, AZ 85724, USA

**Keywords:** lens epithelium, Na,K-ATPase activity, piezo1, calcium

## Abstract

Lens ion homeostasis depends on Na,K-ATPase and NKCC1. TRPV4 and TRPV1 channels, which are mechanosensitive, play important roles in mechanisms that regulate the activity of these transporters. Here, we examined another mechanosensitive channel, piezo1, which is also expressed in the lens. The purpose of the study was to examine piezo1 function. Recognizing that activation of TRPV4 and TRPV1 causes changes in lens ion transport mechanisms, we carried out studies to determine whether piezo1 activation changes either Na,K-ATPase-mediated or NKCC1-mediated ion transport. We also examined channel function of piezo1 by measuring calcium entry. Rb uptake was measured as an index of inwardly directed potassium transport by intact mouse lenses. Intracellular calcium concentration was measured in Fura-2 loaded cells by a ratiometric imaging technique. Piezo1 immunolocalization was most evident in the lens epithelium. Potassium (Rb) uptake was increased in intact lenses as well as in cultured lens epithelium exposed to Yoda1, a piezo1 agonist. The majority of Rb uptake is Na,K-ATPase-dependent, although there also is a significant NKCC-dependent component. In the presence of ouabain, an Na,K-ATPase inhibitor, Yoda1 did not increase Rb uptake. In contrast, Yoda1 increased Rb uptake to a similar degree in the presence or absence of 1 µM bumetanide, an NKCC inhibitor. The Rb uptake response to Yoda1 was inhibited by the selective piezo1 antagonist GsMTx4, and also by the nonselective antagonists ruthenium red and gadolinium. In parallel studies, Yoda1 was observed to increase cytoplasmic calcium concentration in cells loaded with Fura-2. The calcium response to Yoda1 was abolished by gadolinium or ruthenium red. The calcium and Rb uptake responses to Yoda1 were absent in calcium-free bathing solution, consistent with calcium entry when piezo1 is activated. Taken together, these findings point to stimulation of Na,K-ATPase, but not NKCC, when piezo1 is activated. Na,K-ATPase is the principal mechanism responsible for ion and water homeostasis in the lens. The functional role of lens piezo1 is a topic for further study.

## 1. Introduction

The ocular lens is a multicellular structure specialized for transparency [1]. Mature fibers, which constitute the bulk of the lens, are highly differentiated, tightly packed, ribbon-like cells that lack nuclei, mitochondria, and other organelles. A monolayer of undifferentiated epithelial cells covers the anterior lens surface. Extensive cell–cell coupling enables the cell mass to function more or less as a syncytium [2]. Ion and water homeostasis depends, to a large extent, on transport mechanisms in the epithelial monolayer [3]. Several studies point to a role for feedback loop mechanisms that regulate the activity of Na,K-ATPase or NKCC1 in the epithelium in a way that apparently corrects for swelling or shrinkage of the structure. TRPV4 and TRPV1 channels have been found in the lens, and it has been proposed that these ion channels, which are mechanosensitive, operate in the feedback loops as sensors activated by distortion, which is caused by small changes in intracellular hydrostatic pressure in cells at the lens surface [4,5]. In a recent study on mouse retinas, Morozumi and coworkers detected a different mechanosensitive channel, piezo1, not only in retinas, but also in the corneal epithelium and lens epithelium [6]. Piezo1 is expressed in a wide variety of tissues, and its global knockout is lethal [7] although it can be deleted in a tissue-specific manner, for example, in urothelium or regulatory T cells [8,9]. Piezo1 is a mechanically gated cation channel that is capable of sensing membrane tension. Due to its bowl-like structure, piezo1 is thought to influence, and be influenced by, membrane curvature [10]. Cells utilize mechanosensitive ion channels as a means of transducing mechanical stimuli into ionic or electrical signals [11]. The homotrimeric piezo1 channel is permeable to monovalent cations Na^+^ and K^+^, as well as divalent cations Ca^2+^ and Mg^2+^ [12]. It inactivates rapidly in a voltage-dependent manner [13]. 

The presence of piezo1 in the lens raises questions regarding its role. The purpose of the present study was to examine piezo1 function in mouse lens. Recognizing that activation of TRPV4 and TRPV1 causes changes in lens ion transport mechanisms, we carried out studies to determine whether piezo1 activation changes either Na,K-ATPase-mediated or NKCC1-mediated ion transport. We also examined the channel function of piezo1 by measuring calcium entry.

## 2. Results

Piezo1 expression in the lens was examined by immunolocalization. Piezo1 was evident as a green stain in the lens epithelium, which covers the anterior lens surface, but was not detectable in fiber cells (Figure 1A). The negative control, in which no primary antibody was used, showed no green staining for piezo1 (Figure 1B). A hematoxylin and eosin-stained eye section is included to show the anatomical orientation of the lens with respect to other eye tissues (Figure 1C). Primary cultured lens epithelium also showed rich expression of piezo1 (Figure 2).

Because the epithelium is known to play a critical role in lens Na-K homeostasis, lenses were exposed to the piezo1 agonist Yoda1, while Rb uptake was measured as an indicator of inwardly directed potassium transport. Previous studies have validated Rb uptake as an indicator of inward potassium transport by Na,K-ATPase and NKCC in the lens [14,15] and many other tissues [16,17,18]. Intact lenses exposed to 1 µM Yoda1 responded with a 40–50% increase in the rate of Rb uptake (Figure 3A). Because studies on piezo channels in the urinary tract revealed certain differences in the response of males and females [9], the Rb uptake response was measured separately in lenses obtained from male and female mice. Yoda1 exposure increased Rb uptake to the same degree in male and female lenses (Figure 3B).

In order to reduce animal usage, further studies on piezo1 responses were carried out on male mouse lens epithelium in primary culture. When the influence of Yoda1 on cultured cells was examined over a range of concentrations, maximal stimulation of Rb uptake was observed at ~1 µM Yoda1 (Figure 4A). There was no difference in the control and Yoda1-induced Rb uptake responses in cultured lens epithelium from male or female mice (Figure 4B).

Previous lens studies have indicated that the majority of Rb uptake is Na,K-ATPase-dependent, although there also is a significant NKCC-dependent component. In order to study the Rb uptake response in greater detail, cells were exposed to Yoda1 in the presence or absence of the selective Na,K-ATPase inhibitor ouabain. Mouse Na,K-ATPase is known to be ouabain-resistant in comparison to other species, and initial studies revealed that the full effect of ouabain on Rb uptake required a concentration of 1 mM (Figure 5 inset). In the presence of 1 mM ouabain, Yoda1 did not increase Rb uptake (Figure 5A). The results point to an effect of Yoda1 on Na,K-ATPase-mediated Rb uptake. In the presence of 1 µM Yoda1, the ouabain-inhibitable component of potassium (Rb) uptake was 0.27 ± 0.02 µmoles/mg protein/10 min (*n* = 6), compared to a value of 0.15 ± 0.01 (*n* = 6) without Yoda1 (Figure 5B).

On the other hand, Yoda1 increased Rb uptake to a similar degree in the presence or absence of maximal inhibitory concentration of bumetanide (1 µM), an NKCC inhibitor (Figure 6A). The results suggest there is no effect of Yoda1 on NKCC-mediated Rb uptake. The bumetanide-inhibitable component of potassium (Rb) uptake, which is only a small portion of the total uptake, was 0.05 ± 0.02 (*n* = 6) µmoles/mg protein/10 min in the presence of Yoda1, compared to 0.03 ± 0.01 (*n* = 6) without Yoda1 (Figure 6B).

Previous studies reported the blockade of piezo1 by ruthenium red and gadolinium [19,20,21]. When cells were exposed to Yoda1 in the presence of 50 µM ruthenium red, the Yoda1-induced increased Rb uptake response was abolished (Figure 7A). Experiments with gadolinium were carried out in a bicarbonate-free HEPES buffer in order to avoid interactions with bicarbonate ions which would reduce free gadolinium concentration [22]. In the presence of 50 µM gadolinium, Yoda1 did not increase Rb uptake (Figure 7B). Importantly, the piezo1 antagonist GsMTx4 (5 μM) also significantly inhibited Yoda1-induced Rb uptake (Figure 7C).

Yoda1 was observed to increase cytoplasmic calcium concentration in cells loaded with Fura2 (Figure 8A). Experiments were also conducted to test the viability of the cells after Yoda1 treatment. In these experiments, Yoda1-treated cells were subsequently exposed to 1 µM ionomycin. Ionomycin caused a further increase in intracellular calcium concentration (Figure 8B). The calcium response to Yoda1 was almost abolished by the stretch-activated channel blockers ruthenium red and gadolinium (Figure 8C,D). In order to test whether the calcium response involved piezo1-mediated entry of extracellular calcium, cells were exposed to Yoda1 in a nominally calcium-free bathing solution. Under these conditions, the Yoda1-induced calcium rise was almost eliminated, but the increase was robust when calcium was introduced to the bathing solution (Figure 8E). Studies were carried out to test whether the increase in Rb uptake observed in cells exposed to Yoda1 required the entry of extracellular calcium. This was the case. Yoda1 did not increase Rb uptake in cells exposed to a nominally calcium-free bathing solution (Figure 9).

## 3. Discussion

Piezo1 protein was detected in the epithelial monolayer that covers the anterior surface of the mouse lens, and intact lenses displayed rapid functional responses to Yoda1, a selective Piezo1 agonist. The findings point to the expression of operational piezo1 channels in the lens. Yoda1 increased Rb uptake by the intact lens, and ruthenium red and gadolinium, two blockers of stretch activated channels, prevented the response. The Rb uptake response to Yoda1 was also prevented by the spider venom GsMTx4, an inhibitor that appears more selective for piezo channels, piezo1, and piezo2 [23,24,25]. Earlier studies confirmed that GsMTx4 did not inhibit TRPV4 activity, whereas HC067047, a specific TRPV4 inhibitor, almost completely blocked the TRPV4 activity [23]. Because transport mechanisms and potassium channels generally carry Rb ions in a manner similar to potassium ions [16,17,18], the Rb uptake response to Yoda1 points to an increase in inwardly directed potassium ion transport. Na,K-ATPase and Na/K/2Cl cotransport (NKCC) are the two principal mechanisms responsible for potassium transport into cells. Studies on cultured lens epithelium revealed that the response to Yoda1 was abolished by ouabain, which is a potent and highly selective Na,K-ATPase inhibitor [26]. In contrast, the Rb uptake response to Yoda1 was not prevented by bumetanide, a widely used selective NKCC inhibitor [27]. Taken together, the results indicate that the increase in Rb uptake stemmed from Yoda1-induced increase in the activity of Na,K-ATPase, and not from NKCC-mediated transport. Obviously, we cannot rule out piezo1-mediated effects on other transporters, but there is no evidence to suggest this is the case, as the Yoda1 response was eliminated by ouabain, the Na,K-ATPase inhibitor. Despite reports of sex differences between certain piezo1 responses [9], the Rb uptake responses to Yoda1 were identical in lenses and cells obtained from male and female mice.

Piezo1 cation channels are understood to be relatively nonselective [12], and channels in the open state could be conduits for Na^+^, K^+^, and Rb^+^ ions. Because Rb^+^ ions behave similarly to K^+^ ions, the electrochemical driving force would tend to drive outward channel-mediated diffusion. On the other hand, the electrochemical driving force favors Na^+^ entry. We considered two possible mechanistic explanations for the observed increase in Na,K-ATPase-mediated inward potassium transport when piezo1 channels are activated. Increased Na^+^ entry via piezo1 channels might cause a rise in cytoplasmic Na^+^ concentration, which contributes to the stimulation of active Na-K transport in lenses and lens cells exposed to Yoda1. Alternatively, increased Ca^2+^ entry via piezo1 channels might activate signaling pathways that stimulate Na,K-ATPase activity. The calcium entry mechanism seems more likely because the Rb uptake response to Yoda1 was abolished in calcium-free medium. The findings are consistent with stimulation of Na,K-ATPase activity which is caused by the rise in cytoplasmic calcium concentration that occurs when Ca^2+^ ions enter via piezo1 channels. There was nothing in the data that hinted of Rb uptake via open piezo channels.

Stretch-activated channel blockers ruthenium red and gadolinium inhibited Rb uptake and calcium response to Yoda-1. Ruthenium red itself produced a small transient calcium response in lens epithelium. We cannot answer with confidence why ruthenium red caused the increase. However, ruthenium red has widespread effects on subcellular calcium handling, and could potentially affect the release of calcium from mitochondria or the endoplasmic reticulum [28]. Gadolinium, on the other hand, has a small effect on the basal level of Rb uptake. This may be because it can inhibit other channels, including certain TRP channels [29]. Ca^2+^ ion entry via piezo1 channels appears to cause an increase in Na,K-ATPase-mediated ion transport. In previous studies, we have shown an ATP-induced rise in cytoplasmic calcium concentration leads to increased Na,K-ATPase activity by a mechanism that involves Src kinase activation [30,31,32]. However, the influence of calcium signaling on Na,K-ATPase is complex, because elevated cytoplasmic calcium caused by certain stimuli, for example, endothelin-1, can lead to a reduction in Na,K-ATPase activity [33]. The mechanistic link between piezo1-mediated calcium entry and increased Na,K-ATPase activity remains to be determined. The signaling pathways activated by calcium entry might be complex. In porcine lens epithelium, calcium entry is known to cause cAMP to increase due to the action of a calcium-activated adenylyl cyclase [34].

Piezo1-associated calcium entry was confirmed by exposing Fura2-loaded lens cells to Yoda1, which was found to cause a significant rise in cytoplasmic calcium concentration. Consistent with the notion of calcium entry, the response was absent in cells exposed to Yoda1 in a calcium-free medium. Importantly, the calcium response was also prevented by ruthenium red and gadolinium, both of which are known to block stretch-activated ion channels. Recently, mechanical stretch has been determined to activate piezo1 in caveolae of alveolar type 1 cells [35]. Interestingly, stretch-activated Ca^2+^ entry via piezo1 was shown to cause activation of pannexin-1 hemichannels and ATP release from the alveolar epithelium [35]. Such ATP release is critical to the mechanism of stretch-activated surfactant secretion during lung inflation, which is essential to pulmonary function. In the lens, we previously reported hypoosmotic stress-induced ATP release [36,37].

It seems likely that piezo1 serves a functional role as a stretch-activated channel in the lens. Feasibly, piezo1 might sense and respond to osmotic swelling or changes in the tension exerted on the lens by the zonules (suspensory ligaments). Piezo1 is not the only stretch-activated channel expressed by lens cells. TRPV4 and TRPV1 play critical roles in feedback loops pathways that regulate the activity of ion transporters in the lens epithelium [5]. Lens TRPV4 responds to osmotic swelling [38], while TRPV1 responds to osmotic shrinkage [15], so in this respect, the two channels appear to act in functional opposition. It remains to be determined whether functional piezo1 responses overlap to any degree with TRPV4- and TRPV1-mediated responses. However, it is noteworthy that piezo1 activation and TRPV4 activation both cause an increase in Na,K-ATPase-mediated active transport. In studies on chondrocytes, it has been suggested that TRPV4 and piezo1 respond to different strength stimuli, with TRPV4 responding to a lower degree of strain than piezo channels [39,40]. Studies on urothelial cells revealed something different: the sensitivity of Piezo1 for stretch stimulation was higher than that of TRPV4, suggesting piezo1 is a more sensitive mechanosensor than TRPV4 [23]. Crosstalk between calcium regulatory proteins and mechanosensitive ion channels, including TRP and piezo channels, appears to be important in cardiovascular pathophysiology [41].

Previous investigations observed that cyclic stretching of alveolar epithelial cells increases Rb uptake and causes a rise in the amount of Na,K-ATPase α1 catalytic subunit protein at the basolateral membrane [42]. The Rb uptake response was absent in cells treated with Gd^3+^ in order to block stretch-activated ion channels. Piezo1 was not known at that time, and the authors suggested that the Na,K-ATPase response stemmed from an increase in sodium entry in cells subjected to stretching. More recently, mechanical stretch has been determined to activate piezo1 in the caveolae of alveolar type 1 cells [35]. In vascular smooth muscle cells, which express two Na,K-ATPase catalytic subunit isoforms, α1 and α2, a stretch stimulus increased the abundance of both isoforms, but gadolinium prevented only the α2 isoform response [43].

To be clear, piezo1 expression in the lens is not unexpected. Piezo1 is expressed in many different tissues, and its activation has been associated with a wide variety of responses in the vascular system, lungs, urinary tract, red blood cells [44], and even tumor metastasis [45]. The channel might have a different function in different tissues. It has been suggested that the spatial distribution of individual piezo1 channels may influence the response [10]. Recently, it has been reported that piezo1-mediated calcium entry into the lens, caused by an extended period (24 h) of Yoda1 exposure, stimulates phosphorylation of myosin light chain kinase and activates calpain [46]. Our present study focused on short-term responses of the lens epithelium, as these play a critical and dynamic role in ion homeostasis for the entire lens structure. The piezo1 expression pattern merits further study. While we observed piezo1 immunofluorescence only in the epithelium, others have reported expression in the lens fibers [46]. Elsewhere in the eye, piezo1 has been described in cells of the aqueous humor outflow pathway. There is evidence that functional responses to piezo1 activation include increased matrix metalloproteinase expression, release of PGE2, and altered trabecular meshwork cell proliferation, suggesting that piezo1 might have a role in intraocular pressure regulation [47,48].

## 4. Materials and Methods

### 4.1. Chemicals and Antibodies

Rubidium chloride (RbCl) and other chemicals were purchased from Sigma (St. Louis, MO, USA) and Fura2-AM was purchased from Thermo Fisher Scientific (Waltham, MA, USA). Rabbit polyclonal anti-piezo1 antibody was purchased from Protein Tech Group (Cat. No.: 15939-1-AP, Rosemount, IL, USA). Goat anti-Rabbit IgG (H + L) Alexa Fluor Plus 488 secondary antibody was purchased from Thermo Fisher Scientific (Catalog # 35552, Waltham, MA, USA). DAPI used for nuclear counterstaining was obtained from Thermo Fisher Scientific (Waltham, MA, USA).

### 4.2. Krebs Solution

Bicarbonate-buffered Krebs solution that contained (in mM) 119 NaCl, 4.7 KCl, 1.2 KH_2_PO_4_, 25 NaHCO_3_, 2.0 CaCl_2_, 1 MgCl_2_, and 5.5 glucose was used for most of the experiments. In experiments using gadolinium, a HEPES buffered Krebs solution that contained (in mM) 113 NaCl, 4.7 KCl, 25 HEPES-Na salt, 2.0 CaCl_2_, 1 MgCl_2_, and 5.5 glucose was used. Rubidium uptake experiments were carried out using rubidium-containing Krebs solutions, in which KCl and KH_2_PO_4_ were replaced by adding an equivalent concentration of RbCl and NaH_2_PO_4_, respectively. The pH for all solutions was adjusted to 7.4. Bicarbonate buffered Krebs solution was bubbled with 5% CO_2_ and 95% air for 45 min before pH adjustment to 7.4. HEPES buffered Krebs solution was not bubbled with CO_2_/air.

### 4.3. Lenses and Cultured Lens Epithelium

Lenses were obtained from adult (18–20 weeks) male and female wild type C57BL/6J mice (Jackson Laboratory, Bar Harbor, ME, USA). The use of animals was approved by the Institutional Animal Care and Use Committee (IACUC) of the University of Arizona. The approved protocol number for mouse lens experiments is 18–492. Following CO_2_ euthanasia, eyes were removed, and each lens was isolated by careful dissection and transferred to Krebs solution. Intact lenses were used in some studies. Other studies were carried out using cultured lens cells obtained from male mice, unless otherwise specified. As described earlier [5,36], the capsule and attached lens epithelium was isolated from each lens, and 4–6 tissue samples were placed on a culture dish (60 mm) in a CO_2_ incubator at 37 °C. Then, 0.5 mL of complete culture medium was placed around the border. After 30 min, 3–4 mL of complete medium was added to flood the dish and cover the epithelium explants. The complete medium was prepared using an Epithelial Cell Medium kit (Sciencell Research Laboratories, Carlsbad, CA, USA): 500 mL Basal Epithelial Cell Medium, 10 mL Fetal Bovine Serum (FBS), and 5 mL of a mixture of penicillin and streptomycin. The medium was changed at day 3 and then on alternate days. When enough cells had grown out of the explants, which takes 7–8 days, the cells were trypsinized and propagated as follows. The medium was removed and the cells were washed 2× with Ca^2+^free, Mg^2+^ free HBSS and then subjected to low speed shaking for 3 min in 4.0 mL of 0.25% Trypsin EDTA solution. An equal volume of a mixture of FBS and newborn calf serum (1:1) was added to neutralize the trypsin, then the cell suspension was centrifuged at 167× *g* for 10 min. The pellet was then resuspended in 4 mL of complete medium and seeded in 25 cm^2^ flask at a density of ~10,000–15,000 cells/cm^2^. The medium was changed after 1 day, then on alternate days. Cells typically became confluent within 4–5 days and were propagated to the next passage. The studies reported here used 3rd and 4th passage cells.

### 4.4. Rubidium Uptake by Intact Lens

The measurement of rubidium (Rb^+^) uptake by intact lenses and cultured lens cells has been described earlier [14]. Rb uptake is established as a selective, accurate, and reproducible indicator of inward potassium transport by Na,K-ATPase and NKCC [16,17,18]. Because Rb uptake by the lens is linear for at least 25 min [14], the amount of Rb accumulated during a 10 min uptake period indicates the uptake rate. In the lens, as in most tissues, Na,K-ATPase and NKCC are the two principal mechanisms responsible for potassium (Rb) uptake. Each component can be examined separately, using ouabain to inhibit the Na,K-ATPase-mediated component and bumetanide to inhibit the NKCC-mediated component [15]. Freshly isolated intact lenses were immersed in 8.0 mL Krebs solution in a 6-well plate at 37 °C in a CO_2_ incubator. Freshly isolated lenses were allowed to recover for 3 h before use, then were carefully transferred to the Krebs solution containing test agents. For Rb uptake, lenses were placed in RbCl-containing Krebs solution for 10 min. At the end of the Rb uptake period, lenses were washed briefly with ice-cold isotonic (100 mM) MgCl_2_ solution, containing 2.0 mM BaCl_2_. Then, each lens was blotted gently on moistened filter paper, placed in a 75 mm test tube, and weighed. The lenses were then dried at 60 °C for 7 days and weighed again for water content determination by weight loss. Dried lenses were then digested in 200 µL 30% nitric acid in a water bath at 55 °C for 24 h. The digest was diluted with 1.8 mL of double distilled water and centrifuged at 1507× *g* for 15 min. The supernatant was used to measure Rb using an atomic absorption spectrophotometer (AAnalyst 100, Perkin Elmer, Waltham, MA, USA). Rb uptake was expressed as mmoles/kg lens dry weight. Rb uptake was compared in intact lenses obtained either from male or female animals.

### 4.5. Rb uptake by Lens Epithelium

Cells grown on 6-well plates were washed 2 times using prewarmed (37 °C) Krebs solution, then incubated in the normal Krebs solution for at least 1 h before starting the Rb uptake protocol. Rb uptake was initiated by replacing the normal Krebs solution with Rb-containing Krebs solution for 10 min. After this, the lens or cells were washed twice by immersion in ice-cold isotonic MgCl_2_ solution. Rb accumulation was determined by digesting the tissue in 30% nitic acid, diluting the solution as necessary with deionized water, and measuring Rb concentration using an atomic absorption spectrophotometer (Perkin Elmer Analyst 100).

### 4.6. Cytoplasmic Calcium

Cytoplasmic calcium was measured in cells loaded with Fura-2 AM by recording fluorescence intensity at alternating excitation wavelengths of 340 nm and 380 nm, and the emission collected at 510 nm, using an imaging system purchased from Intracellular Imaging Inc. (Cincinnati, OH, USA). As described previously (Mandal, Shahidullah et al. 2015), sparely seeded nonconfluent cells (~70% confluent, 48 h in culture) plated on a 35 mm culture dish were loaded by incubating in 1 mL of normal Krebs solution, containing 10 μM Fura-2 AM, for 25 min at 37 °C. Fura-2 AM solution (1 mM) was prepared in DMSO with 20% pluronic acid. At the end of the loading period, the cells were washed 5 times with normal Krebs solution to remove extracellular Fura-2 AM. Then the dish was mounted on an open perfusion mini-incubator (Harvard Apparatus, Model PDMI-2) attached to the stage of an inverted microscope (Nikon Eclipse TS100). The cells were superperfused (1.5 mL per min) with normal Krebs’ solution equilibrated with 95% air/5% CO_2_ to obtain a baseline. The superfusion was switched to a Krebs solution containing the test compound. The temperature of the perfusate was maintained at 37 °C using a temperature controller (Harvard Apparatus, Holliston, MA, USA, Model TC. 202A). The fluorescence ratios were converted into free calcium concentration by the system-integrated software, with a calibration curve obtained according to the manufacturer’s protocol, using a fura-2 calcium calibration kit supplied by Invitrogen. In each experiment, data from 15 to 30 individual cells were averaged and considered as *n* = 1.

### 4.7. Immunolocalization on Intact Eye Section

Immunolocalization studies were carried out on intact eye paraffin sections using an approach described earlier [30,37]. Mouse eyes were fixed in 10% neutral buffered formalin (NBF) for 48–72 h at 2–4 °C. A needle puncture was made through the optic nerve to facilitate formalin penetration. Then, the eyes were transferred to 70% ethanol at 2–4 °C for at least 24 h or until preparation of the paraffin blocks by a standard procedure. Next, 5-micron-thick sections were cut and deparaffinized.

Tissue sections were incubated at room temperature for 90 min in 10% goat serum in PBS (blocking buffer). A primary antibody directed against piezo1 was added (1:100 dilution of the supplied 500 µg/mL stock solution) and the samples were incubated for 24 h at 4 °C. Control specimens received no primary antibody, but only the blocking buffer. The specimens were washed 3 times with PBS and incubated for 90 min at room temperature with a fluorescent secondary antibody (Alexa Fluor 488, 1:200 dilution, in blocking buffer). Nuclear counterstaining was carried out by incubating with DAPI (1:100 in PBS, 300 nM) for 10 min at room temperature. Stained samples were covered with a scanty amount of Dako Antifade Mounting Medium (Agilent Technologies, Inc., Santa Clara, CA, USA), and a cover slip was placed carefully over the sample, avoiding any bubble formation. Images were captured using a Leica DMI 6000 microscope (Leica Microsystems, Deerfield, IL, USA). Fluorescence excitation was achieved using 488 and 358 nm laser excitation wavelengths for Alexa Fluor 488 and DAPI, respectively.

### 4.8. Immunolocalization on Cultured Lens Epithelium

For immunolocalization studies on cultured lens epithelium, cells grown on specially designed chamber slides (Nalgene Nunc, Lab-Tek II Chamber Slides), were washed with phosphate buffered saline (PBS) containing 1.0 mM MgCl_2_ and 0.1 mM CaCl_2_, then fixed in acetone for 2 min at room temperature. The fixed cells were incubated at room temperature for 90 min in 10% goat serum in PBS (blocking buffer). Rabbit polyclonal primary antibody directed against Piezo1 was added (1:100 dilution of the supplied 500 µg/mL stock) to the cells and incubated for 24 h at 2–4 °C. Control specimens received only the blocking buffer. The cells were washed with PBS and incubated for 90 min at room temperature with a fluorescent secondary antibody (Alexa Fluor 488, 1:200 dilution in blocking buffer). Nuclear counterstaining was carried out by incubating with DAPI (1:100 in PBS, final concentration 300 nM) for 10 min at room temperature. Images were captured using a Leica DMI 6000 microscope (Leica Microsystems, Deerfield, IL, USA). Fluorescence excitation was achieved using 488 and 358 nm laser excitation wavelengths for Alexa Fluor and DAPI, respectively.

### 4.9. Statistical Analysis

A two-sample t test was used to compare the difference between the two groups of data. One-way analysis of variance (ANOVA), followed by Šídák’s multiple comparison tests, was used to compare differences for more than two groups of data. A probability (*p*) value of <0.05 was considered significant.

## 5. Conclusions

These findings point to stimulation of Na,K-ATPase, but not NKCC, when piezo1 is activated in the lens. Calcium entry through the piezo1 channel is responsible for the stimulation of Na,K-ATPase activity. Since Na,K-ATPase is the principal mechanism responsible for ion and water homeostasis in the lens, the functional role of lens piezo1 is an important topic for further study.

## Figures and Tables

**Figure 1 ijms-23-12870-f001:**
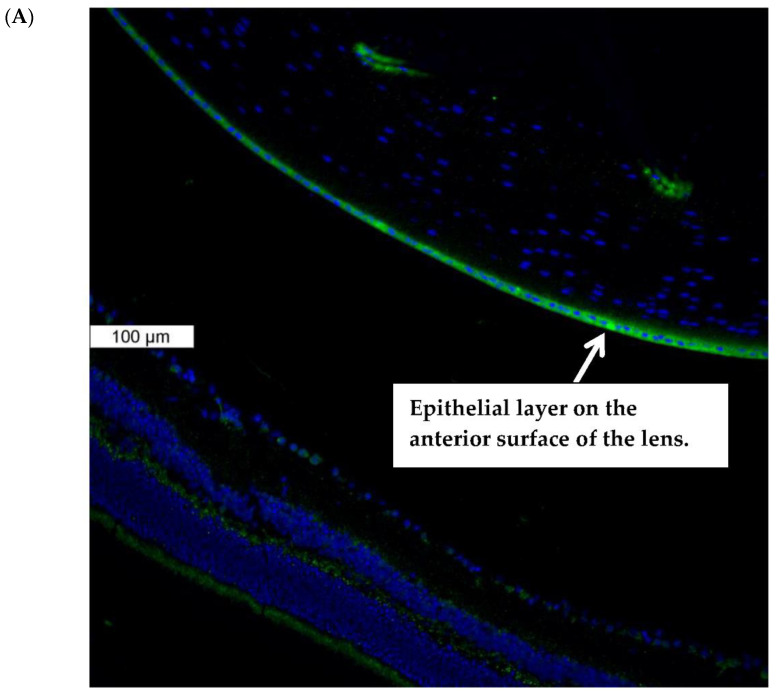
Immunolocalization of piezo1 in mouse lens. (**Panel A**) shows piezo1 (green) and nuclei (blue) in the epithelial cell monolayer, which covers the anterior surface of the lens. The image shows the equatorial region of the lens, where nuclei are evident in the differentiating fiber cells of the outer lens cortex. (**Panel B**) shows the negative control, in which no primary antibody was used. (**Panel C**) shows a section of the eye, stained with Hematoxylin and Eosin, that shows the lens with respect to the iris, ciliary body, retina, and cornea. The epithelial cell layer, which plays a critical role in ion transport, is positioned at the anterior lens surface, but not at the posterior surface.

**Figure 2 ijms-23-12870-f002:**
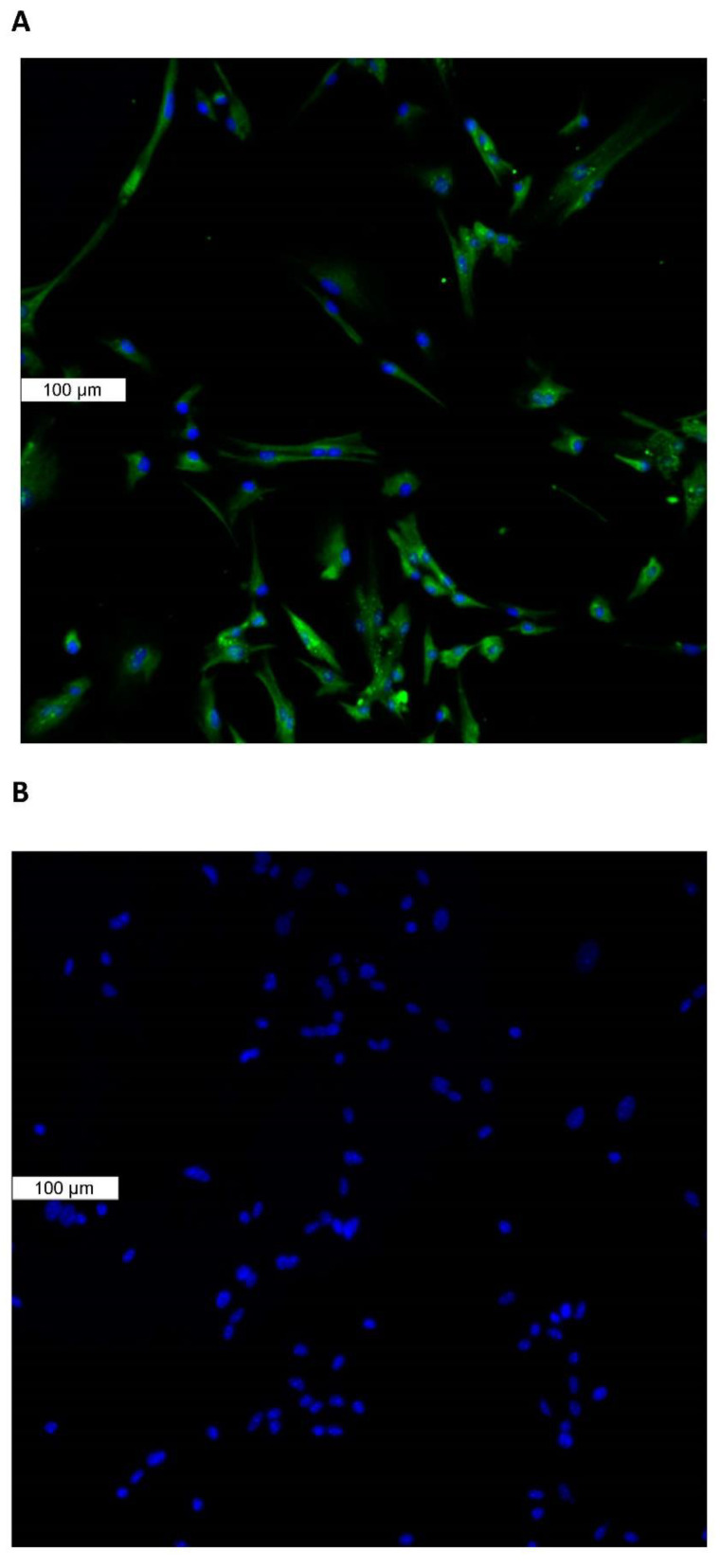
Immunolocalization of piezo1 in primary cultured mouse lens epithelium. (**Panel A**) shows piezo1 (green) and nuclei (blue). (**Panel B**) shows the negative control, in which no primary antibody was used.

**Figure 3 ijms-23-12870-f003:**
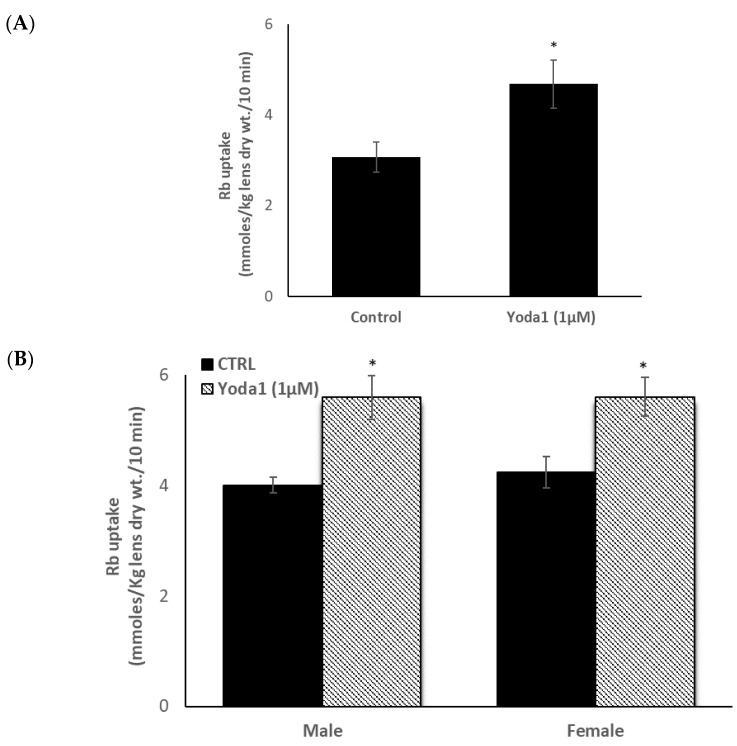
The influence of Yoda1 on rubidium (Rb) uptake by intact male mouse lenses. Lenses were incubated for 10 min in Rb-containing Krebs solution + 1 µM Yoda1 (**Panel A**). In a separate experiment, Rb uptake was compared between lenses from male and female mice (**Panel B**). The values are the mean ± SEM of results from 5–7 lenses. * Indicates a significant difference (*p* < 0.05) from the control, as determined by two sample *t*-tests.

**Figure 4 ijms-23-12870-f004:**
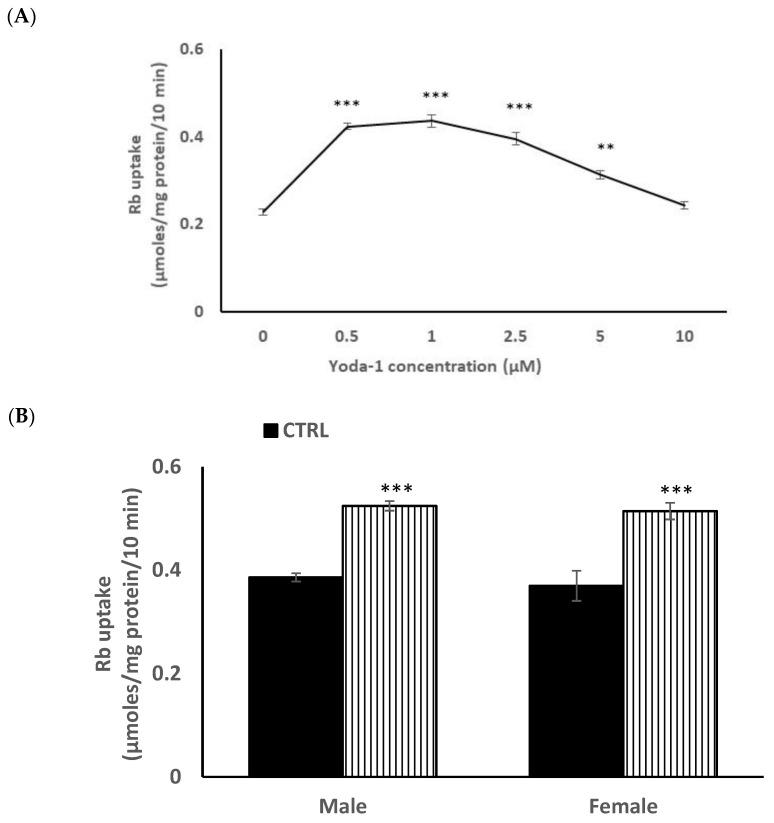
Concentration-dependence of the Yoda1 effect on Rb uptake in primary cultured mouse lens epithelium (**Panel A**). Cells were exposed to Yoda1 in the range 0.5–10 µM for 10 min in Rb-containing Krebs solution. In a different experiment, the Rb uptake response to 1 µM Yoda1 was measured separately in cultured epithelium derived from male mice or from female mice. (**Panel B**) The values are the mean ± SEM of results from 6 independent experiments. *** and ** indicatesignificant differences (*p* < 0.001 and 0.01, respectively) from the control, as determined by one-way ANOVA followed by Šídák’s multiple comparison test.

**Figure 5 ijms-23-12870-f005:**
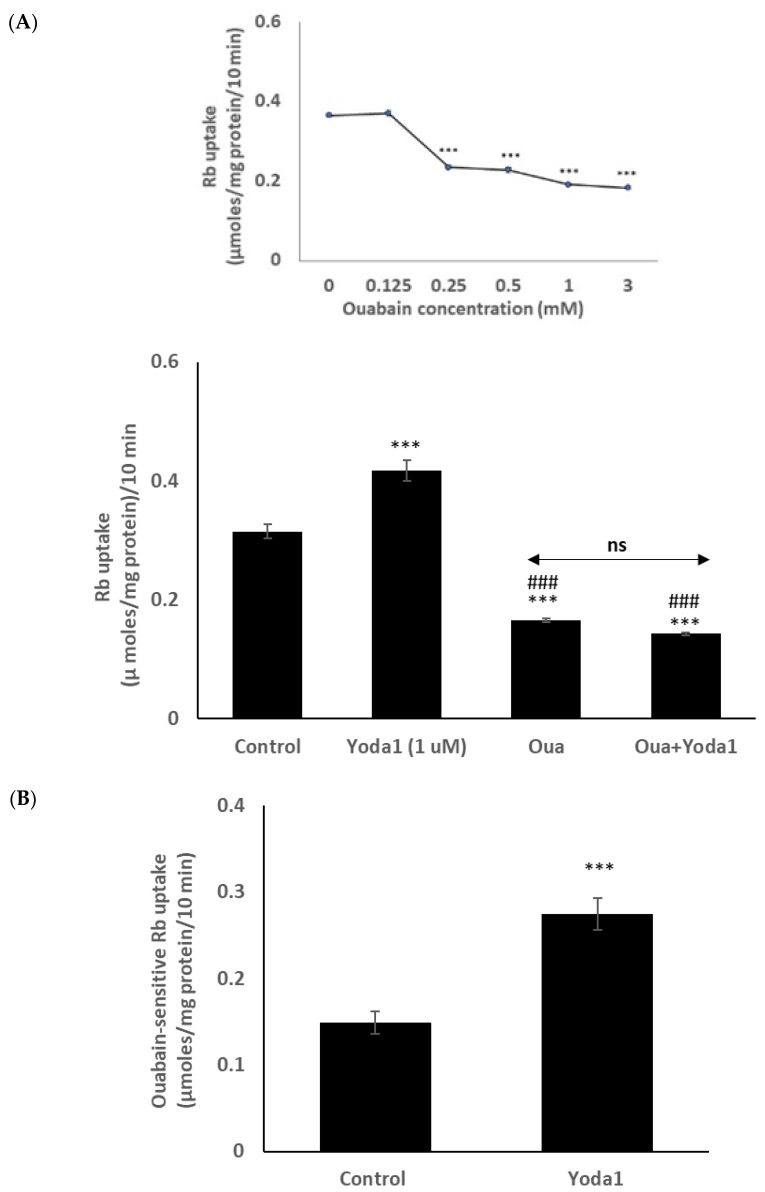
In the presence of the Na,K-ATPase inhibitor ouabain (1 mM), Yoda1 did not increase the Rb in cultured lens epithelium. (**Panel A**) shows Rb uptake in cells that were exposed to Yoda1 (1 µM) in Rb-containing Krebs solution ± ouabain for 10 min. Added alone, ouabain reduced Rb uptake. The concentration-dependence of the ouabain effect on Rb uptake is shown in the upper left (inset). (**Panel B**) shows the ouabain-sensitive component of Rb uptake. The values are the mean ± SEM of results from 6 independent experiments. *** indicates a significant difference (*p* < 0.001) from the control, and ^###^ indicates a significant difference (*p* < 0.001) from Yoda1 treatment, as determined by one-way ANOVA followed by Šídák’s multiple comparison test. ns = not significant.

**Figure 6 ijms-23-12870-f006:**
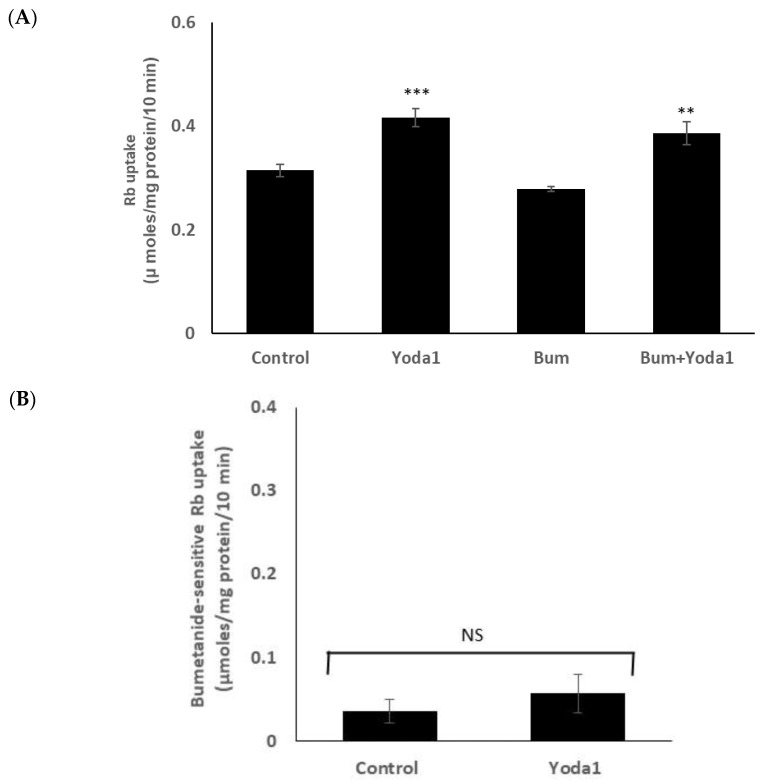
Yoda1 increased Rb uptake to a similar degree in the presence or absence (control) of the NKCC inhibitor bumetanide (1 µM). (**Panel A**) shows Rb uptake in cells exposed to Yoda1 (1 µM) in Rb-containing Krebs solution ± bumetanide for 10 min. Added alone, bumetanide slightly reduced Rb uptake, but not to a significant degree. (**Panel B**) shows the bumetanide-sensitive component of Rb uptake. The values are the mean ± SEM of results from 6 independent experiments. ** and *** indicate significant differences (*p* < 0.01 and *p* < 0.001, respectively) from the control (no Yoda1), as determined by one-way ANOVA followed by Šídák’s multiple comparison test. NS = not significant.

**Figure 7 ijms-23-12870-f007:**
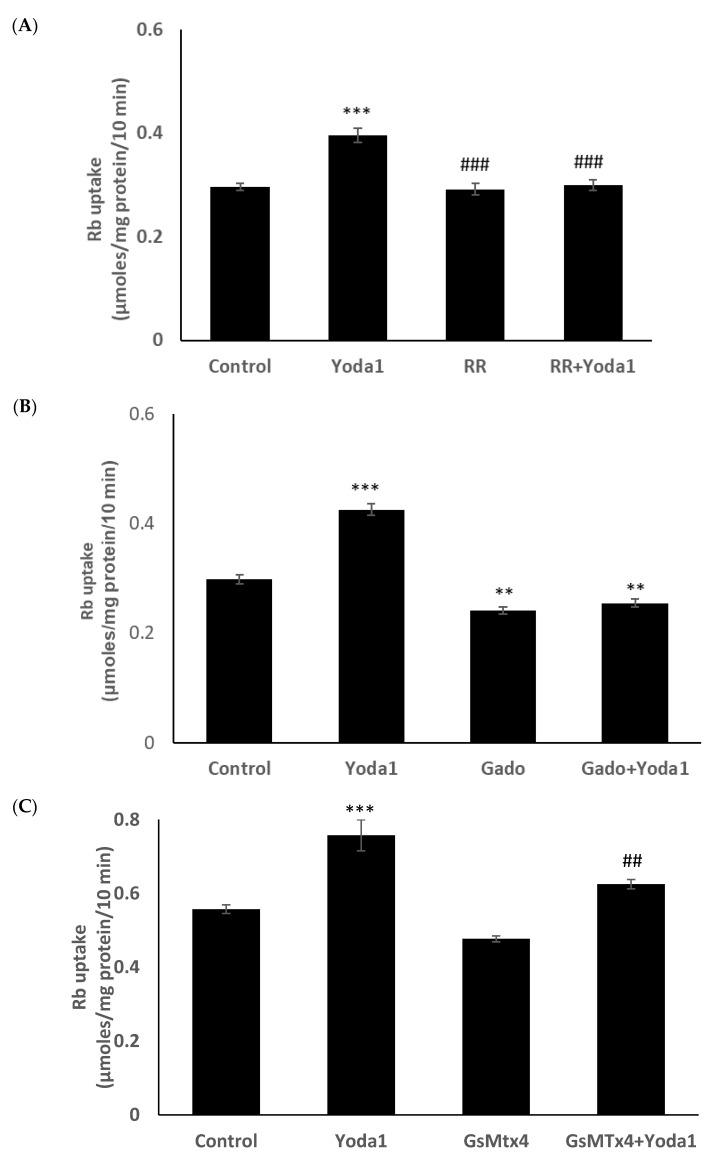
The influence of stretch-activated channel inhibitors ruthenium red (**A**) and gadolinium (**B**), and a selective piezo channel inhibitor GsMTx4 (**C**) on the Rb uptake response to Yoda1 in cultured lens epithelium. Cells were exposed to Yoda1 (1 µM) in Rb-containing Krebs solution ± ruthenium red (50 μM) for 10 min. Some cells were exposed to Yoda1 (1 µM) in Rb-containing, bicarbonate-free, HEPES-buffered Krebs solution ± gadolinium (50 µM) for 10 min. Another group of cells was exposed to Yoda1 (1 µM) in Rb-containing Krebs solution ± GsMTx4 (5 µM) for 10 min. The values are the mean ± SEM of results from 6 independent experiments. *** and ** indicate significant differences (*p* < 0.001 and *p* < 0.01, respectively) from the control (no Yoda1); ^###^ and ^##^ indicate significant differences (*p* < 0.001 and *p* < 0.01, respectively) from Yoda1 treatment as determined by one-way ANOVA followed by Šídák’s multiple comparison test.

**Figure 8 ijms-23-12870-f008:**
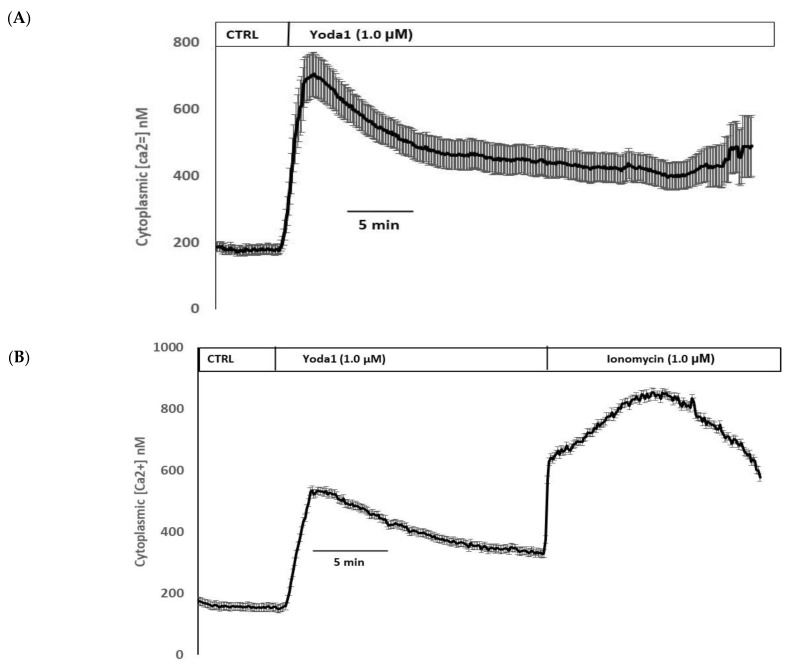
(**A**) The cytoplasmic calcium response to Yoda1 in control conditions, (**B**) subsequent exposure to ionomycin, (**C**) in the presence of ruthenium red or (**D**) gadolinium, (**E**) in a calcium-free solution. Cultured lens epithelial cells loaded with Fura2 were first superfused with a control Krebs solution, then Yoda 1 (1 µM) was introduced. Some cells received ruthenium red (50 µM) before Yoda 1 was added in the continued presence of ruthenium red. Gadolinium (50 µM) experiments were carried out in a similar manner, although a bicarbonate-free HEPES-buffered Krebs solution was used. The values shown are the mean ± SEM of results from 6–10 independent experiments, where each individual experiment was the average of 15–30 individual cells.

**Figure 9 ijms-23-12870-f009:**
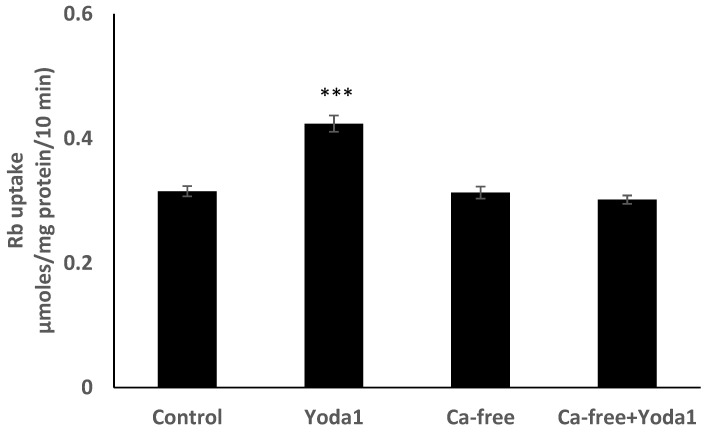
In a calcium-free bathing solution, Yoda1 did not increase the Rb in cultured lens epithelium. Cells were exposed to Yoda1 (1 µM) for 10 min in either a calcium-free Krebs solution, or in a normal Krebs solution. The values are the mean ± SEM of results from 6 independent experiments. *** indicates a significant difference (*p* < 0.001) from the control, as determined by one-way ANOVA followed by Šídák’s multiple comparison test.

## Data Availability

Raw data are available on request.

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
