# Peer review of "Activation of Piezo1 Increases Na,K-ATPase-Mediated Ion Transport in Mouse Lens"

_ijms, 2022, doi:10.3390/ijms232112870_

Round 1

Reviewer 1 Report

This manuscript investigated whether Piezo1 activation by chemical agonist would interact with Na,K-ATPase-mediated transporter. Although the study is of potentially interest, it does not fully convince the reviewer the message the authors want to convey. 

Specific concerns and comments:

1.       The authors only relied on Rb uptake measurement throughout the manuscript. It is unclear how reliable this measurement is. This needs to be strongly justified in the introduction. Potassium entry could take place through many different routes on the membrane.

2.  The authors attribute the Rb uptake solely to Na,K-ATPase-mediated ion transport. Piezo1 is a relatively novel ion channel with many unknown. If Rb uptake takes place through Piezo1 directly, all the current conclusions would be re-drawn. The authors need to rule out this possibility.

 3.       If the cross-talk between Piezo1 and Na,K-ATPase-mediated ion transport does take place, there is no any mechanistic investigation in the current version, which makes this manuscript particular weak to be considered for publication in this journal at this stage. The authors should extend the research by investigating how both proteins communicate with each other.

 4.       The introduction is not clearly laid out and understood, especially for Piezo1 importance in this type of cells or similar type of cells. The relevant literatures should be added in the references: (Int. J. Mol. Sci. 202122(16), 8782; Cell Rep. 2020 Oct 6;33(1):108225. )

 5. In figure 1: there is no any scale bar shown. In addition, it is also unclear how representative these two figures would be. Statistical figure or description would be helpful in clarifying this point.

6. After the cells were primarily cultured, the concentration of Rb uptake was reduced from millimolar to micromolar (over 10,000 time difference). This result seems extremely odd. 

7. In figure 2, the increase of Rb uptake by 1 micromolar Yoda1 is roughly by 50%, while that is roughly over 100% in figure 3. Both results do not match with each other.

8. In figure 2A, the concentration in A is below the average of either male or female in B. One wonders how reliable the result shown in figure A would be in comparison to figure B.   

9. The labelling in figure 3 is not properly done.  

10.  Again, in figure 3, the concentration for control in figure A is far higher than that in figure B. This discrepancy needs to be explained.

11. Currently there is no any study carried out to rule out that Yoda1 does not any effect on transporter, so there could be a chance that Yoda1 might exert an effect on transporter. The authors need to clarify this.

12.   Nearly almost all experiments were carried out with pharmacological approach. The specificity of these drugs or compounds applied is therefore a major concern. This needs to be thoroughly addressed.

Author Response

PDF file has been attached containing point-by-point responses to reviewer's comments.

Reviewer 2 Report

Dear Authors,

In this manuscript, the authors investigated the role of Piezo1 in potassium influx of lens epithelial cell culture. The authors used Yoda1 as Piezo1 activator, and ruthenium red or Gad3+ as non-specific inhibitor for stretch-activated channels. They concluded that Ca2+ influx through Piezo1 was involved in ion homeostasis in lens epithelial cells, especially modulating Na+,K+-ATPase-dependent K+ influx.

However, the reviewer could not imagine the actual role of Piezo1 from this manuscript, because they did not show detail localization of Piezo1 proteins in their immunohistochemistry (e.g., apical or basolateral). They did not mention about actual stimulus for activation for Piezo1; they just used an activator (Yoda1) but they did not mention anything about actual physiological stimulus for the opening of Piezo1 in lens epithelium. Moreover, they haven’t done any Piezo1 specific inhibition.

Major comments;

1)     Introduction part: the purpose of this study should be indicated.

2)     Methods: The approved protocol number of animal experiments should be indicated.

3)     Fig. 1: The antibody used in Fig. 1 was not very suitable for the immunohistochemistry in mouse tissue/cells. The reviewer believes that this is against human Piezo1 proteins. Antibody absorption experiment using mouse Piezo1 proteins should be done to show its specificity. The dilution rate should also be indicated in the method part.

4)     Fig. 1: Please add scale bars. Also, higher resolution images including the information of apical/basolateral membrane (e.g., co-staining with Na+-K+-ATPase) should be done. Bright field images (HE staining) should be added as well because the reviewer did not get the orientation of the lens tissue from current version of Fig. 1.

5)     Fig. 2-9: The immunocytochemistry should be done to show which cell types were used for Fig. 2-9. Smooth muscle actin (SMA) or other fibroblast markers might be helpful. Moreover, it might be better if they can confirm Piezo1 expression in their cell culture. A change in total lens volume, cell volume, and the cellular localization of alpha2 isoform of Na+,K+-ATPase during the treatment of Yoda1, might support the authors hypothesis. A bright-field image of cultured cells should be added as well, with a scale bar.

6)     Fig. 2-9: Regarding the inhibition of Piezo1, any specific method should be done (e.g., siRNA knockdown, tissue from cKO mouse, or more specific inhibitor such as GsMTx4).

7)     Fig. 8: Ionomycin treatment should be done at the end of each experiment to show cell viability.

8)     Discussion part: Actual physiological stimulus for Piezo1 opening should be discussed. What is the physiological role of Piezo1 in lens epithelial cells in vivo?

Minors;

1)     Materials and Methods: “Thermo Fisher Scientific (Waltham, MA)”, font size should be corrected.

2)     Fig. 2: The unit (mmoles/kg lens dry wt./10 min) should be re-calculated into “micromoles/mg protein/10 min”.

3)     Fig. 2: The authors just mentioned “intact lenses were used in some studies.” please indicate experimental procedure of Fig. 2.

4)     Fig. 4: Legend: “Yoda1 did not increase the Rb uptake in cultured lens epithelium in the presence of the Na+,K+-ATPase inhibitor ouabain (1 mM).”

5)     Fig. 5: Bumetanide-sensitive Rb uptake should be presented a graph with completely the same scale as the panel B of Fig. 4 (not like 0.05, 0.1), to show these values in Fig. 5 were relatively small.

6)     P8: “Previous studies report blockade of Piezo1 by…” => “Previous studies reported a blockade of Piezo1…” Also, please cite appropriate papers here.

7)     Fig. 6 and 7: They can be combined.

8)     Fig. 7: Please explain why the Rb uptake with Gd3+ was smaller compared to control.

9)     Fig. 8: Please explain why the intracellular Ca2+ level transiently increased by a ruthenium red treatment.

10)  P15, please spell out “IOP”.

Sincerely,

Author Response

PDF file has been attached with point-by-point responses to reviewer's comments.

Round 2

Reviewer 1 Report

The authors have addressed my concerns and questions. 

Author Response

  1. Spell check has been performed throughout the manuscript in response to comments by both reviewers (minor spell check required).

Reviewer 2 Report

Dear Authors,

The reviewer has just a minor comment below.

1)Abstract: "The calcium response to Yoda1 was markedly inhibited by GsMTx4" : data should be presented as a figure, or omit this sentence.

Sincerely,

Author Response

  1. Spell check has been performed throughout the manuscript in response to comments by both reviewers (minor spell check required).
  2. As suggested by the reviewer, we deleted a line having a typographical error in the Abstract.